# Victimization status among persons with disabilities and its predictors: Evidence from Bangladesh National Survey on Persons with Disabilities

**Mizanur Rahman**[1], **Md Shohel Rana**[2,3], **Gulam Khandaker**[4,5,6], **Md Mostafizur Rahman**[7], **Md. Nuruzzaman Khan**[2,8]*

**1** Department of Fisheries, Jamalpur Science & Technology University, Jamalpur, Bangladesh, **2** Department of Population Science, Jatiya Kabi Kazi Nazrul Islam University, Mymensingh, Bangladesh, **3** Maternal and Child Health Division, International Centre for Diarrhoeal Disease Research, Mohakhali, Dhaka, Bangladesh, **4** CSF Global, Dhaka, Bangladesh, **5** Central Queensland Public Health Unit, Central Queensland Hospital and Health Service, Rockhampton, Australia, **6** School of Health, Medical and Applied Sciences, Central Queensland University, Rockhampton, Australia, **7** Department of Population Science and Human Resource Development, University of Rajshahi, Bangladesh, **8** Nossal Institute for Global Health, Melbourne School of Population and Global Health, The University of Melbourne, Melbourne, Australia

* sumonrupop@gmail.com

## Abstract

### Background

Persons with disabilities often face various forms of victimization, yet there is limited research exploring this phenomenon in Bangladesh. This study aims to investigate the victimization status among persons with disabilities and identify its predictors.

### Methods

Data of 4293 persons with disabilities analyzed in this study were extracted from the 2021 National Survey on Persons with Disabilities. Victimization status (yes, no) was considered as the outcome variables. Explanatory variables considered were factors at the individual, household, and community levels. A multilevel mixed-effect logistic regression model was used to explore the association of the outcome variable with explanatory variables by dividing the total sample into age groups of 0–17 years, 18–59 years, and ≥60 years.

### Results

The study found that 44% of persons with disabilities in Bangladesh experienced victimization, primarily involving neighbours (90.64%), relatives (43.41%), friends (28.41%), and family members (27.07%). Among persons aged 0–17 years, increasing age was associated with a higher likelihood of being victimized, while residing in the wealthiest households or in certain divisions like Khulna and Rangpur was associated with lower likelihoods. Conversely, among respondents aged 18–59 and ≥60 years, increasing age was associated with a lower likelihood of being victimized. Unmarried respondents

**Data availability statement:** We analyzed secondary data available at the Bangladesh Bureau of Statistics (BBS) after submitting a research proposal. While we are not authorized to share the data file directly, we have ensured all relevant data for replication are included in the tables. Interested researchers can access the raw data file by submitting a research proposal to the BBS through the following link: https://bbs.gov.bd/, following the same process we used.

**Funding:** The author(s) received no specific funding for this work.

**Competing interests:** The authors have declared that no competing interests exist.

**Abbreviations:** PWD: Persons with Disabilities, WHO: World Health Organization, LMICs: Low- and Middle-Income Countries, BBS: Bangladesh Bureau of Statistics, aOR: Adjusted Odds Ratios, AIC: Akaike Information Criteria, BIC: Bayesian Information Criteria.

aged 18–59 years had an increased likelihood of victimization compared to married individuals. For persons with disabilities aged ≥60 years, a higher level of schooling was associated with a reduced likelihood of being victimized. We also found divisional differences in the likelihood of victimization, with respondents residing in Chattogram, Khulna, Mymensingh, Rangpur, and Sylhet reporting lower likelihoods compared to those residing in the Barishal division, among persons with disabilities aged 18–59 and ≥60 years.

## Conclusion

This study's findings underscore that around 4 in 10 persons with disabilities are being victimized. Tailored programs and awareness-building initiatives covering neighbours, relatives, friends, and family members of persons with disabilities are important to ensure dignified lives for this population.

## Introduction

As of 2022, an estimated 1.3 billion people worldwide were living with disabilities, constituting the largest minority group [1]. More than 80% of these individuals reside in low- and middle-income countries (LMICs), with projections indicating an increase in their numbers due to improvements in healthcare facilities and medical technology, leading to the survival of individuals who may have otherwise succumbed to their conditions [2,3]. Additionally, the rising incidence of road traffic injuries contributes significantly to the burden of disability, particularly in LMICs [4]. Bangladesh, a LMICs with a Gross National Income (GNI) of $2,864 and the eighth largest population in the world, approaching 168 million, plays a significant role in this global trend. This is due to substantial advancements in healthcare facilities and medical technology, as well as a rise in road traffic injuries [5,6]. As of 2022, Bangladesh is home to over four and a half million persons with disabilities, accounting for 2.79% of the total population [7,8]. Consequently, the reduction of disability through comprehensive healthcare services has become a global priority.

Persons with disabilities in worldwide encounter various challenges [9–12]. This poses a particular challenge for LMICs, as challenges arise from multiple levels. The primary level of challenges arises from community misconceptions, such as the belief that disability is primarily caused by parental curses [9,10]. Other misconceptions include viewing persons with disabilities as burdens on society and as dependent groups. These existing community-level challenges in LMICs, including high unemployment rates and an education system that lacks disability-friendly provisions, can exacerbate these issues by impeding the ability of persons with disabilities to engage with mainstream society [8]. As a result, persons with disabilities in LMICs often rely on social safety net programs operated by governmental and non-governmental organizations or on support from family members to meet their basic needs, including food, education, and healthcare [13].

However, the support received under these programs is often insufficient to enable them to live with dignity. For example, in Bangladesh, persons with disabilities receive approximately 8 USD per month under government-operated social safety net programs, which is inadequate to fulfil even single basic requirements [13]. Consequently, they frequently report poor health conditions compared to the general population, with these conditions often left untreated due to the lack of disability-friendly healthcare facilities—a situation prevalent in Bangladesh and other LMICs [14–17].

Furthermore, in addition to community-level misconceptions about disability, persons with disabilities are often victimized by members of their own community and even by family members [18–20]. This victimization further increases their vulnerability in society, impacting their participation in available support programs despite their existence [20]. Victims may hesitate to seek help due to fear, shame, or distrust, and the psychological effects of victimization can undermine their confidence and agency [21]. However, despite these pathways, the extent of victimization remains largely unexplored in LMICs, including Bangladesh. Existing studies in these settings predominantly focus on the prevalence of disability, determinants of disability, health conditions of persons with disabilities, access to healthcare services, and coverage of social safety net programs [13–15,17,22]. Studies on victimization and its determinants are largely unexplored in LMICs. Our comprehensive search found only a few studies, and they are limited in scope, focusing primarily on women and children or analysing small samples [23–25]. These studies provide evidence of a higher prevalence of victimization, particularly among women and children with disabilities—a pattern that differs from findings in studies conducted in high-income countries [26–28]. However, studies focusing on victimization among persons with disabilities are completely lacking in Bangladesh. Therefore, we conducted this study to explore the extent of victimization among persons with disabilities in Bangladesh, as well as the factors associated with it.

## Methods

### Sampling strategy

We analyzed data extracted from the National Survey on Persons with Disabilities (NSPD), a nationally representative household survey conducted in 2021 by the government of Bangladesh. A two-stage stratified random sampling technique was employed to select the households. Initially, 800 primary sample units (PSUs) were chosen from a list of 293,579 PSUs generated by the Bangladesh Bureau of Statistics during the 2011 National Population Census, which was the most recent census available at the time of the survey. Household listing operations were then carried out in each selected PSU. Subsequently, in the second stage of sampling, 45 households were systematically selected from each chosen PSU, resulting in a list of 36,000 households, from which data were collected from 35,493 households, yielding a response rate of 98.6%. All 155,025 respondents from these selected households were included in the survey. Data were collected through face-to-face interview through structured questionnaire. A detailed explanation of the sampling procedure and the collected data has been published elsewhere [29,30].

### Analytical sample

Of the survey sample, 4,293 respondents reported having a disability, comprising 2.79% of the total sample. This subset of respondents was analyzed in alignment with the study's objectives. The inclusion criteria utilized to derive this subset were as follows: (i) individuals self-reporting disabilities, and (ii) those who responded to questions concerning experiences of discrimination or harassment by individuals or groups on various grounds within 12 months preceding the survey.

### Outcome variables

The outcome variable under consideration was the experience of discrimination or harassment among persons with disabilities on various grounds. Each person with a disability was queried during the survey: "*In the past 12 months, have you personally felt discriminated*

*against or harassed based on the following grounds?*" They were presented with multiple options, including disability, ethnicity/immigration, sex, age, religion/belief, and other reasons. Participants (or their caregivers/parents for persons with disability aged < 18 years) were instructed to indicate "yes," "no," or "don't know" for each applicable option. Subsequently, we reclassified these responses into a single dichotomous variable for victimization status—whether individuals experienced discrimination or harassment based on their characteristics—categorized as "Yes" (indicating a positive response to any of the provided options) or "No" (reflecting negative or uncertain responses across all options).

## Explanatory variables

The selection of explanatory variables in this study followed a two-stage process. Initially, we conducted a thorough search across various databases using pertinent keywords, focusing on Bangladesh and LMICs [13–15,17,22,23,26–28]. Variables identified in relevant studies were compiled into a list. Subsequently, these listed variables were cross-referenced with the survey data we analyzed, resulting in a refined list of selected and available variables for consideration in this study. These then categorised under three broad themes (individual-level factors, household level factors, and community level factors) as per the socio-ecological model of health [31]. Individual level factors encompassed respondent's age (0–17 years, 18–59 years, and ≥ 60 years), years of schooling (treated as continuous variable), gender (male or female), occupation (agriculture, blue-collar work, pink-collar work, white-collar work, student, housewife, unable to work, and others), and marital status (married, unmarried, widowed/divorced/separated). Household-level variable considered was household wealth quintile (poorest, poorer, middle, richer, richest) and religion (Islam, others). Household wealth quintile variable was derived by the survey authority through principal component analysis of household asset-related variables such as roofing type and ownership of a refrigerator. Furthermore, additional factors taken into account were respondents' place of residence (urban or rural) and their region of residence (Barishal, Chattogram, Dhaka, Khulna, Mymensingh, Rajshahi, Rangpur, and Sylhet) and they categorised under community level variables.

## Statistical analysis

Descriptive statistics were employed to characterize the respondents analyzed in this study. The statistical significance of victimization status across the considered explanatory variables was assessed using a chi-square test. To investigate the relationship between victimization status and the explanatory variables, a multilevel mixed-effect logistic regression model was utilized. This choice was motivated by the nested structure of the NSPD data, where respondents were nested within households, and households were nested within a PSU. Previous studies have demonstrated that multilevel modelling provides more accurate results than conventional simple logistic regression models when dealing with such nested data structures. Three distinct models were run by categorizing the total sample into three age groups: 0–17 years, 18–59 years, and ≥ 60 years. This division was made considering the inclusion of individuals with disabilities in national-level policies and programs based on these age categories. A progressive model-building approach was adopted for each age category, comprising four distinct models. Model 1 served as the null model, considering only victimization status. In Model 2, individual-level factors were incorporated along with victimization status, while both household- and individual-level factors were included in Model 3. Model 4 encompassed all individual, household, and community-level factors. Prior to running each model, multicollinearity was assessed using the Variance Inflation Factor (VIF). If evidence of multicollinearity was found (VIF > 5.0), the relevant variable was removed, and the model was rerun.

The results were reported as adjusted odds ratios (aOR) along with their corresponding 95% Confidence Intervals (95% CI). All statistical analyses were performed using STATA/SE 14.0 (Stata Corp LP, College Station, Texas, United States of America).

### Ethics approval

The survey data we analyzed received ethical approval from the Ethics Committee of the Bangladesh Bureau of Statistics (BBS). Informed written consent was obtained from all participants or their legal guardians in cases where participants were minors (under 18 years old). We accessed the data from BBS by submitting a specific research proposal for this study. Before sharing the data, BBS de-identified it by removing all respondent-identifying variables. Since the data was de-identified, no further ethical approval was required.

## Results

### Background characteristics of the respondents

The background characteristics of the respondents are summarized in Table 1. The mean age of the respondents was 41.44 years. More than half (51%) of the total respondents fell into the age bracket of 18–59 years at the time of the survey. Approximately 59% of the respondents were male. Nearly one-third of the total respondents identified themselves as unable to work. Around 80% of the respondents resided in rural areas, while approximately 22% indicated Dhaka as their region of residence.

### Victimization status and basis and extent of victimization across types of disability in Bangladesh

The overall prevalence of victimization status and its breakdown across several bases of victimization are presented in Table 2. We found 43.73% of the total respondents reported experiencing victimization within 12 months of the survey, with the primary basis of victimization being disability itself (43.20%). A majority of the persons with disabilities reported single basis for their victimization.

Persons with intellectual disabilities reported higher prevalence of victimization (66%) following by autism spectrum disorder (63%), mental disabilities (62%) and down syndrome (59%) (Fig 1).

### Persons or groups by whom respondents with disabilities were victimized within 12 months of the survey

We explored the individuals or groups responsible for victimizing persons with disabilities, and the results are presented in Fig 2. We found that neighbours (91%) constituted the primary group victimizing persons with disabilities, followed by relatives (43%), friends (28%), and family members (27%).

### Victimization status of persons with disabilities across individual, households and community level factors

Table 3 presents the d\istribution of victimization status across individual, household, and community-level factors. Children aged 0–17 years, respondents in other occupation categories, unmarried individuals, and respondents residing in the Barishal division reported a higher prevalence of being victimized within 12 months of the survey. We found significant differences in victimization status across individual, household, and community-level factors. Age-specific prevalence of victimization are presented in S1 Table on S1 File. We found a

**Table 1. Socio-demographic characteristics of the respondent (N = 4,293).**

| Characteristics | Frequency (n) | Percentage (%) |
|---|---|---|
| **Individual level factor** | | |
| **Respondent's age in years,** mean (±SD) | | 41.44 (±23.59) |
| 0–17 | 892 | 20.78 |
| 18–59 | 2,190 | 51.00 |
| ≥ 60 | 1,211 | 28.22 |
| **Respondent's gender** | | |
| Male | 2,513 | 58.55 |
| Female | 1,780 | 41.45 |
| **Respondents year of schooling,** mean (SD) | | 3.89(±9.01) |
| **Respondent's occupation** | | |
| Agriculture | 411 | 9.58 |
| Blue collar worker[b] | 290 | 6.76 |
| Pink collar worker[p] | 165 | 3.83 |
| White collar worker[w] | 375 | 8.75 |
| Student | 484 | 11.27 |
| Housewives | 505 | 11.76 |
| Unable to work | 1,413 | 32.92 |
| Others* | 650 | 15.15 |
| **Respondent's marital Status** | | |
| Married | 2,062 | 48.03 |
| Unmarried | 1,505 | 35.05 |
| Widow/Divorce/Separate | 726 | 16.92 |
| **Household level factor** | | |
| **Religion** | | |
| Muslim | 3,843 | 89.52 |
| Others** | 450 | 10.48 |
| **Wealth Quintile** | | |
| Poorest | 1,164 | 27.12 |
| Poorer | 942 | 21.95 |
| Middle | 853 | 19.87 |
| Richer | 727 | 16.92 |
| Richest | 607 | 14.14 |
| **Community level factor** | | |
| **Place of residence** | | |
| Rural | 3,470 | 80.83 |
| Urban | 823 | 19.17 |
| **Region of residence** | | |
| Barishal | 227 | 5.29 |
| Chattogram | 697 | 16.22 |
| Dhaka | 923 | 21.50 |
| Khulna | 597 | 13.91 |
| Mymensingh | 311 | 7.24 |
| Rajshahi | 662 | 15.42 |

*(Continued)*

**Table 1.** (Continued)

| Characteristics | Frequency (n) | Percentage (%) |
|---|---|---|
| Rangpur | 633 | 14.74 |
| Sylhet | 243 | 5.67 |

[b]blue collar worker means [factory/manufacturing workers/labour, transportation/ communication workers, day labor (non-agriculture), auto/ cng/ tempo driver, rickshaw driver/ van driver/ boatman, poultry/ animal husbandry for business, fishery or aquaculture and fisherman].

[p]Pink collar worker means [small business (capital up to taka 1000), business (capital over taka 10000), kabiraj/ojha/ spiritual physician, village doctor and homeopathy doctor].

[w]White collar worker means [teacher, lawyer/journalist/doctor/engineer, government employee/officer, private/Ngo employee/ officer, handicraft/cottage industry, weaver/blacksmith/potter/goldsmith/service, imam/ priest, family helper and housemaid/servant].

[*]Family helper, servant, looking for work, unable for work, beggar, no work and not looking for work, and other unnamed occupations;

[**]Hindu, Buddhism, Christianity, etc.

**Table 2.** Victimization status and reasons of victimization among persons with disabilities in Bangladesh, 2021 (N = 4,293).

| Background characteristics | Victimization status | |
|---|---|---|
| **Reasons of victimization** | **Number of persons with disabilities at risk[*]** | **Victimized, n (%)** |
| Ethnic or immigration origin | 4,276 | 63 (1.47) |
| Sex | 4,282 | 52 (1.22) |
| Age | 4,287 | 105 (2.44) |
| Religion or belief | 4,289 | 23 (0.54) |
| Disability | 4,284 | 1,851 (43.20) |
| Other reasons | 4,254 | 70 (1.63) |
| **Number of reported reasons for victimization** | | |
| Single | 4,293 | 1,660 (38.67) |
| Multiple | 4,293 | 217 (5.06) |
| **Overall prevalence of victimization (%, 95% CI)** | | |
| **Victimized** | 43.73 (95% CI, 41.51–45.98) | |
| **Non-victimized** | 56.27 (95% CI, 54.02–58.49) | |

[*]The number determined was based on characteristics specific eligibility criteria.

higher prevalence of victimization among persons with disabilities aged 15 to 44, which declined thereafter with increasing age.

## Factors associated with being victimized by persons with disabilities in Bangladesh

We compared the intra-class correlation (ICC) and variance of the random intercept across each of the four models run for each of three groups (S2, S3, and S4 Tables in S1 File). The best model was identified by the least ICC values. For persons with disabilities aged 0–17 years, the ICC decreased from 38% in the null model to 33% in Model 4. A similar declining trend was observed in the remaining two models, with Model 4 showing the lowest ICC value in each case. Therefore, Model 4 was selected as the best model to consider.

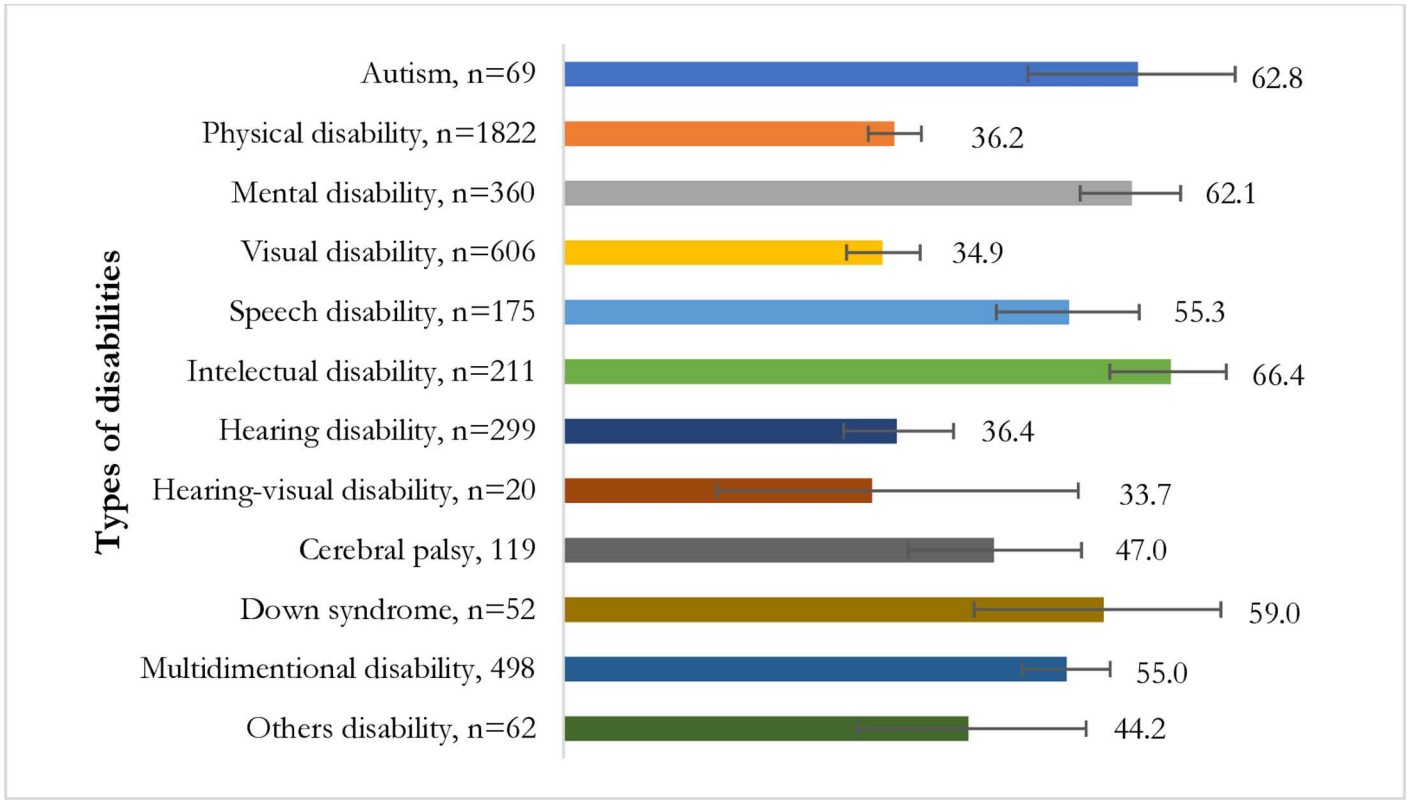

**Fig 1. Victimization status across type of disabilities.**

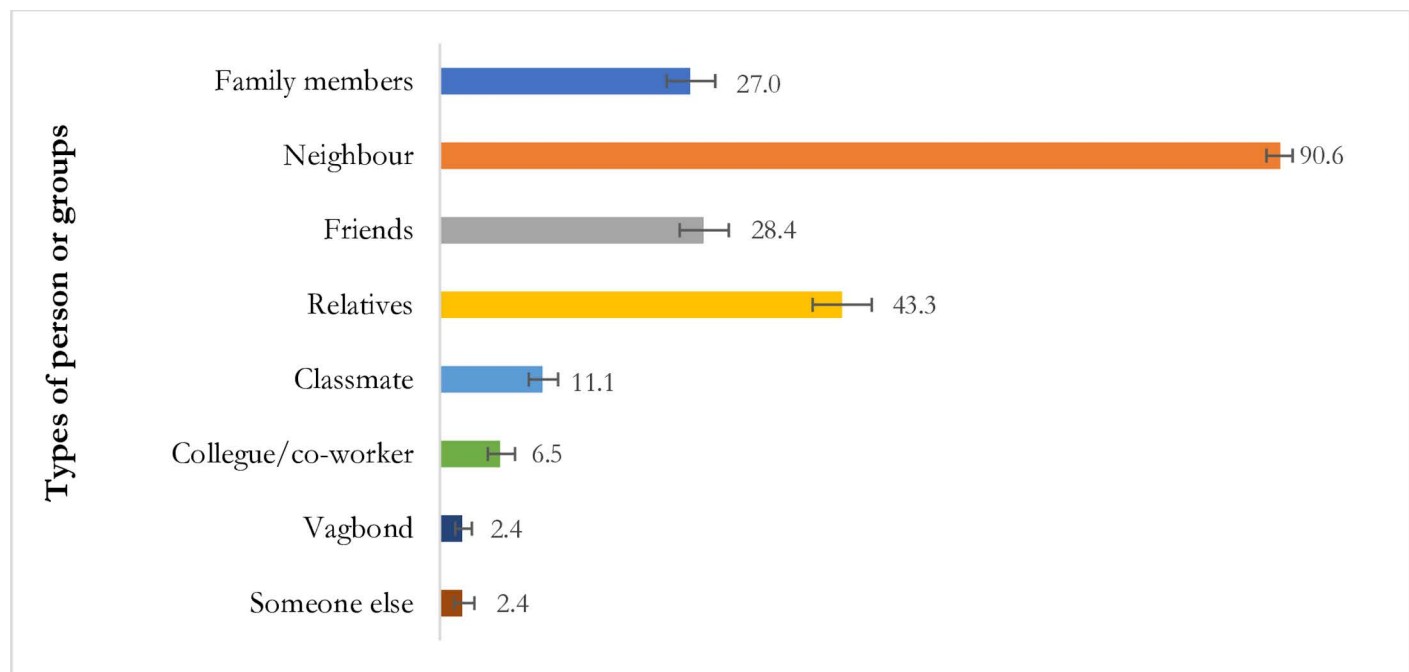

**Fig 2. Persons or groups responsible for victimization of persons with disabilities in Bangladesh, N = 1877.**

**Table 3. Distribution of victimization among persons with disability across individual, household and community level factors.**

| Characteristics | Number of persons with disabilities | Victimized (%) |
|---|---|---|
| **Individual level factor** | | |
| **Respondent's age (in years)** | | |
| 0–17 | 462 | 51.83 |
| 18–59 | 1078 | 49.22 |
| ≥60 | 337 | 27.84 |
| **Respondent's gender** | | |
| Male | 1121 | 44.59 |
| Female | 756 | 42.52 |
| **Respondent's year of schooling** (mean) | 1833 | 3.76 |
| **Respondent's occupation** | | |
| Agriculture | 174 | 42.45 |
| Blue collar worker | 131 | 45.21 |
| Pink collar worker | 65 | 39.4 |
| White collar worker | 158 | 41.99 |
| Student | 238 | 49.29 |
| Housewives | 198 | 39.17 |
| Unable to work | 570 | 40.31 |
| Others | 343 | 52.8 |
| **Marital status** | | |
| Married | 782 | 37.94 |
| Unmarried | 852 | 56.6 |
| Widow/divorced/separated | 243 | 33.5 |
| **Household level factor** | | |
| **Religion** | | |
| Muslim | 1703 | 44.33 |
| Others | 174 | 38.64 |
| **Wealth Quintile** | | |
| Poorest | 538 | 46.16 |
| Poorer | 423 | 44.91 |
| Middle | 361 | 42.35 |
| Richer | 323 | 44.51 |
| Richest | 232 | 38.23 |
| **Community level factor** | | |
| **Place of residence** | | |
| Rural | 1528 | 44.05 |
| Urban | 349 | 42.39 |
| **Region of residence** | | |
| Barishal | 129 | 56.81 |
| Chattogram | 300 | 43.16 |
| Dhaka | 507 | 54.92 |
| Khulna | 206 | 34.5 |
| Mymensingh | 123 | 39.5 |
| Rajshahi | 282 | 42.58 |
| Rangpur | 233 | 36.8 |
| Sylhet | 97 | 39.91 |

Distribution presented in row percentages.

The results of Model 4 for each group are presented in Table 4. S2, S3, and S4 Tables in S1 File provide results for all models for persons with disabilities aged 0–17 years, 18–59 years and older aged ≥60 years, respectively.

For the model concerning persons with disabilities aged 0–17 years, we found that for each year increase in the respondent's age, there was a 1.07 times higher likelihood of being victimized (aOR: 1.07, 95% CI: 1.01–1.12). The likelihood of being victimized was lower among the richest individuals (aOR: 0.42, 95% CI: 0.20–0.88) compared to the poorest. Respondents residing in the Khulna and Rangpur divisions reported lower likelihoods of being victimized compared to those residing in the Barishal division.

For persons with disabilities aged 18–59 years, each year increase in age was associated with a 2% decrease in the likelihood of being victimized (aOR: 0.98, 95% CI: 0.97–0.99). We also observed a lower likelihood of being victimized among students (aOR: 0.42, 95% CI: 0.21–0.82) compared to respondents whose occupation was agriculture. Conversely, a higher likelihood of being victimized was found among unmarried individuals (aOR: 2.24, 95% CI: 1.60–3.14) compared to married persons with disabilities. Lower likelihoods of being victimized were found among the wealthiest individuals (aOR: 0.44, 95% CI: 0.28–0.69) compared to the poorest. Additionally, we observed 48% to 66% lower likelihoods of being victimized among persons with disabilities residing in the Chattogram, Khulna, Mymensingh, Rangpur, and Sylhet divisions compared to those residing in the Barishal division.

Additionally, among persons with disabilities aged 60 and older, each additional year of age was associated with approximately a 3% decrease in the odds of being victimized (aOR: 0.97, 95% CI: 0.95–0.99). We found that administrative divisions such as Mymensingh, Sylhet, Khulna, Rangpur, and Chattogram had lower likelihoods of being victimized compared to those residing in Barishal.

## Discussion

The aim of the study was to explore the extent of victimization among persons with disabilities in Bangladesh and the factors associated with it. We found that approximately 44% of the total persons with disabilities in Bangladesh experienced victimization at least once within 12 months of the survey period with disability itself was reported as major basis of being victimized (43.20%). The majority of persons with disabilities were victimized by neighbours, followed by relatives, friends and family members. The major factors associated with victimization were respondents' age, occupation, marital status, wealth quintile, and region of residence, with different directions of association based on the age of the persons with disabilities. These findings indicate a higher vulnerability to victimization among persons with disabilities in Bangladesh and underscore the need for targeted interventions to protect them.

We reported that approximately 44% of the total persons with disabilities in Bangladesh are being victimized, slightly higher than the prevalence of victimization of 40% among persons with disabilities [32]. This reported prevalence is much higher than the prevalence of victimization among persons with disabilities in LMICs, including Vietnam (approximately 3%) [33] and Burkina Faso (13.9%) [23]. Broad reasons for such a higher prevalence of victimization in Bangladesh could include social stigma and misconceptions, which contribute to discrimination and mistreatment of persons with disabilities [13]. Limited awareness and education about disability rights and inclusion may perpetuate harmful stereotypes and increase vulnerability to victimization [20]. Inadequate support systems and services for persons with disabilities leave them more susceptible to exploitation and abuse [8]. Economic hardship and social inequality faced by persons with disabilities in Bangladesh may increase their vulnerability, as

**Table 4. Factors associated with victimization status among persons with disabilities in Bangladesh.**

| Characteristics | Children aged 0–17 | | Adult aged 18–59 | | Older aged ≥ 60 years | |
|---|---|---|---|---|---|---|
| | aOR | 95% CI | aOR | 95% CI | aOR | 95% CI |
| **Individual level factor** | | | | | | |
| Respondent's age in years | 1.07** | 1.01–1.12 | 0.98*** | 0.97–0.99 | 0.97** | 0.95–0.99 |
| **Gender** | | | | | | |
| Male (ref) | 1.0 | 1.0 | 1.0 | 1.0 | 1.0 | 1.0 |
| Female | 0.85 | 0.58–1.25 | 1.11 | 0.81–1.50 | 0.86 | 0.54–1.38 |
| **Respondent's year of schooling** | 0.99 | 0.98–1.01 | 0.99 | 0.97–1.01 | 0.94* | 0.89–1.00 |
| **Respondent's occupation** | | | | | | |
| Agriculture (ref) | Na | Na | 1.0 | 1.0 | 1.0 | 1.0 |
| Blue collar worker | Na | Na | 0.69 | 0.45–1.05 | 3.28 | 0.97–11.02 |
| Pink collar worker | Na | Na | 0.68 | 0.39–1.17 | 1.15 | 0.42–3.15 |
| White collar worker | Na | Na | 0.74 | 0.47–1.17 | 1.12 | 0.52–2.38 |
| Student | Na | Na | 0.42** | 0.21–0.82 | – | – |
| Housewives | Na | Na | 0.74 | 0.46–1.19 | 0.69 | 0.29–1.62 |
| Unable to work | Na | Na | 1.05 | 0.71–1.57 | 0.95 | 0.53–1.69 |
| Others | Na | Na | 1.08 | 0.68–1.71 | 1.65 | 0.79–3.48 |
| **Marital status** | | | | | | |
| Married (ref) | Na | Na | 1.0 | 1.0 | 1.0 | 1.0 |
| Unmarried | Na | Na | 2.24*** | 1.60–3.14 | 1.52 | 0.35–6.60 |
| Widowed/Divorced/Separated | Na | Na | 1.34 | 0.92–1.92 | 0.91 | 0.58–1.43 |
| **Household level factor** | | | | | | |
| **Religion** | | | | | | |
| Muslim (ref) | 1.0 | 1.0 | 1.0 | 1.0 | 1.0 | 1.0 |
| Others | 1.12 | 0.53–2.38 | 0.68 | 0.46–1.01 | 0.85 | 0.48–1.49 |
| **Wealth quintile** | | | | | | |
| Poorest (ref) | 1.0 | 1.0 | 1.0 | 1.0 | 1.0 | 1.0 |
| Poorer | 0.88 | 0.52–1.51 | 0.86 | 0.63–1.17 | 0.84 | 0.16–0.79 |
| Middle | 0.64 | 0.36–1.15 | 0.88 | 0.64–1.22 | 0.94 | 0.44–2.02 |
| Richer | 0.78 | 0.42–1.46 | 0.74 | 0.52–1.06 | 0.31 | 0.13–0.70 |
| Richest | 0.42** | 0.20–0.88 | 0.44*** | 0.28–0.69 | 0.71 | 0.36–1.38 |
| **Community level factor** | | | | | | |
| **Place of residence** | | | | | | |
| Rural (ref) | 1.0 | 1.0 | 1.0 | 1.0 | 1.0 | 1.0 |
| Urban | 1.04 | 0.56–1.95 | 0.84 | 0.56–1.24 | 0.77 | 0.43–1.36 |
| **Region of residence** | | | | | | |
| Barishal (ref) | 1.0 | 1.0 | 1.0 | 1.0 | 1.0 | 1.0 |
| Chattogram | 0.88 | 0.35–2.23 | 0.52* | 0.29–0.95 | 0.35* | 0.16–0.79 |
| Dhaka | 1.95 | 0.77–4.97 | 1.05 | 0.58–1.91 | 0.94 | 0.44–2.02 |
| Khulna | 0.31** | 0.12–0.85 | 0.42** | 0.23–0.95 | 0.31** | 0.13–0.70 |
| Mymensingh | 0.57 | 0.21–1.57 | 0.47* | 0.24–0.95 | 0.14*** | 0.05–0.38 |
| Rajshahi | 0.63 | 0.24–1.61 | 0.55 | 0.30–1.00 | 0.46 | 0.21–1.01 |
| Rangpur | 0.36** | 0.14–0.92 | 0.36** | 0.20–0.67 | 0.35* | 0.16–0.79 |
| Sylhet | 0.57 | 0.21–1.55 | 0.34** | 0.17–0.67 | 0.22** | 0.08–0.60 |
| **Model summary** | | | | | | |
| Intra-class correlation (ICC), (SD) | 0.33 (±0.09) | | 0.31 (±0.04) | | 0.30 (±0.06) | |
| Variance of the random intercept | 1.26 (0.86–1.85) *** | | 1.21 (1.02–1.43) *** | | 1.20 (0.90–1.59) *** | |

*(Continued)*

**Table 4.** (Continued)

| Characteristics | Children aged 0–17 | | Adult aged 18–59 | | Older aged ≥ 60 years | |
|---|---|---|---|---|---|---|
| | aOR | 95% CI | aOR | 95% CI | aOR | 95% CI |
| Akaike's information criterion (AIC) | 1040.1 | | 2832.94 | | 1362.25 | |
| Bayesian information criterion (BIC) | 1123.92 | | 2986.89 | | 1494.83 | |

Note: *p-value < 0.05, **p-value < 0.01, ***p-value < 0.001, aOR: Adjusted odds ratio, CI: Confidence interval.

they often have limited access to healthcare, education, and employment opportunities [32]. Moreover, structural barriers such as inaccessible infrastructure and transportation further isolate persons with disabilities and exacerbate their risk of victimization [34].

We observed varying likelihoods of victimization among persons with disabilities aged 0–17 years and those aged ≥ 18 years, with individuals aged 45 or older showing a decreased likelihood of being victimized. However, we were unable to validate our findings due to a lack of relevant literature. Broad reasons for such differences in likelihoods could include vulnerability and dependency among younger individuals, who may be more susceptible to victimization due to their reliance on caregivers and limited ability to advocate for themselves [32,35]. In contrast, older individuals may have developed stronger social networks and relationships, providing greater protection against victimization [36]. Developmental factors may also play a role, with younger persons facing unique challenges related to social integration and peer relationships, while older individuals may have developed coping strategies and resilience [8,13]. Additionally, differences in access to support systems, protective factors, and resources may contribute to variations in victimization likelihood among different age groups. Environmental and contextual factors, such as living arrangements and community norms, may further shape patterns of victimization across the lifespan [37]. Overall, these findings underscore the complex interplay of individual, social, and environmental factors in shaping the vulnerability to victimization among persons with disabilities of different ages.

We identified lower likelihoods of victimization among persons with disabilities who were students. Possible reasons for this finding may include increased social integration within structured educational environments, where regular interactions with peers and educators can foster supportive relationships and provide a protective buffer against victimization [35,38]. Moreover, educational institutions often offer tailored support services and resources for students with disabilities, such as counselling and accommodations, which can enhance resilience and coping abilities [35]. Additionally, students may benefit from peer support networks within these settings, where they can connect with others facing similar challenges, receive emotional validation, and access practical advice [39]. Furthermore, the supervision and oversight provided by teachers and staff in educational environments may deter victimization and provide avenues for intervention if incidents occur, creating a sense of safety and security for students with disabilities [19]. Lastly, access to education empowers individuals by equipping them with knowledge, skills, and opportunities for personal and academic growth, enhancing self-confidence, assertiveness, and self-advocacy abilities, and enabling them to assert their rights and resist victimization [35].

We found higher likelihoods of victimization among persons with disabilities who were unmarried, consistent with previous studies in LMICs and Bangladesh [19,32,40]. This finding may be attributed to several reasons. Firstly, unmarried individuals with disabilities may experience greater social isolation compared to their married counterparts, lacking the supportive network that a spouse or family can provide [13]. This isolation can leave them more vulnerable to exploitation and abuse. Secondly, unmarried persons with disabilities may face

economic vulnerability, with limited access to shared financial resources or potential dependence on a single income or government assistance [40]. Economic hardship can increase the risk of victimization as individuals may engage in risky situations to meet their basic needs. Thirdly, unmarried individuals may have limited access to resources and support services available to married individuals, such as housing assistance and healthcare, further exacerbating their vulnerability [13]. Additionally, stigma and discrimination related to their unmarried status may contribute to social exclusion and marginalization, perpetuating their vulnerability within society.

Persons with disabilities residing in the wealthiest households consistently demonstrated lower likelihoods of being victimized. This trend may be attributed to various factors. Firstly, individuals in affluent households often have access to greater financial resources, which can provide them with more opportunities to mitigate risks and protect themselves from victimization [8,23]. Economic stability and security may create a protective buffer against exploitation and abuse. Secondly, wealthier households may offer greater social support networks and resources to individuals with disabilities, fostering a sense of safety and security within their environment [32]. Access to supportive family networks, educational opportunities, and community resources may reduce vulnerability to victimization. Additionally, individuals from affluent backgrounds may have higher levels of education and awareness, enabling them to recognize and address potential threats more effectively [3,21]. Moreover, affluent households may prioritize safety and security measures, such as enhanced home security systems or access to safer neighbourhoods, which can further reduce the likelihood of victimization.

We identified regional-level variations in the likelihood of victimization among persons with disabilities. This variation may be attributed to regional disparities in socio-economic status, as reported in previous studies, with areas experiencing higher poverty rates or economic instability potentially facing greater risks of exploitation and abuse [8,13,15]. Additionally, misconceptions and stigma surrounding disability may vary across regions, influencing the treatment and social inclusion of persons with disabilities. Regions with higher levels of disability awareness and acceptance may provide a more supportive and protective environment for individuals with disabilities, thereby reducing their vulnerability to victimization [13]. Furthermore, differences in education enrolment rates and access to educational opportunities between regions may also contribute to variations in victimization rates [5]. Areas with higher rates of educational attainment and enrolment may foster greater awareness of disability rights and inclusion, leading to lower levels of victimization among persons with disabilities.

The findings of this study have broad policy implications. With evidence showing that around 44% of persons with disabilities in Bangladesh are victimized, this study suggests the need for tailored programs to support this vulnerable group at the community level. These programs may include initiatives to ensure their participation in education and income-generating activities, thereby empowering them economically and socially. Additionally, awareness-building programs targeting neighbours, relatives, friends, and family members—groups from whom higher occurrences of victimization were reported—are crucial. These initiatives should be adapted to the specific geographical settings and socio-economic factors of each region to effectively address the diverse needs and challenges faced by persons with disabilities.

This study possesses several strengths as well as a few limitations. To our knowledge, it represents the first investigation conducted in Bangladesh examining the victimization status among persons with disabilities at the national level and its associated socio-demographic factors. The study includes a comparatively large sample size extracted from a nationally representative survey. Sophisticated statistical methods were used to analysed data, encompassing

a broad range of factors. However, the primary limitations of this study include the analysis of cross-sectional data, which limits our ability to establish causality, as the findings are purely correlational. Additionally, the survey relied on self-reported victimization data, which may introduce the potential for misreporting certain experiences of violence or discrimination. Data were collected through questions posed to the respondents without validation opportunities, demonstrating the possibility of recall bias, although any such bias is likely to be random. Similarly, respondents were directly asked about their experience of any form of victimization, which indicate possibility of underreporting or overreporting due to recall bias, stigma, or a preference for disclosing certain incidents while omitting others, particularly those that might contradict the experiences of their friends and family members. Moreover, aside from the factors adjusted in the model, health and environmental factors may contribute to discrimination against persons with disabilities, underscoring their importance for inclusion in the model. Unfortunately, these data were unavailable in the survey, limiting our ability to consider them. However, these limitations are unlikely to significantly impact the study findings given the analysis of large-scale data. Despite these limitations, the findings of this study will contribute to the development of national-level policies and programs.

## Conclusion

This study reveals that approximately 44% of persons with disabilities in Bangladesh experience victimization, with a significant proportion of incidents occurring at the hands of neighbours, relatives, friends, and family members. However, the likelihood of victimization varies across different demographic factors such as age, occupation, marital status, wealth quintile, and region of residence. These findings underscore the imperative for tailored programs aimed at supporting persons with disabilities to ensure their dignified lives. Additionally, awareness-building programs targeting neighbours, relatives, friends, and family members of persons with disabilities are crucial to fostering a more inclusive and supportive environment. Such initiatives are essential for addressing the vulnerabilities faced by persons with disabilities and promoting their full participation and integration within society.

## Supporting information

**S1 File.** Supplementary Table 1: Age -specific distribution and association with victimization among persons with disabilities in Bangladesh. Supplementary Table 2: Factors associated with victimization status among persons with disabilities aged 0–17 years in Bangladesh. Supplementary Table 3: Factors associated with victimization status among persons with disabilities aged 18–59 years in Bangladesh. Supplementary Table 4: Factors associated with victimization status among persons with disabilities aged ≥60 years in Bangladesh. (DOCX)

## Acknowledgement

The authors thank the Bangladesh Bureau of Statistics (BBS)for granting access to the 2021 National Survey on Persons with Disabilities data.

## Author contributions

**Conceptualization:** Mizanur Rahman, Md. Nuruzzaman Khan.

**Data curation:** Mizanur Rahman.

**Formal analysis:** Mizanur Rahman, Md Shohel Rana.

**Software:** Gulam Khandaker.

**Supervision:** Md Mostafizur Rahman, Md. Nuruzzaman Khan.

**Validation:** Md Shohel Rana, Gulam Khandaker, Md Mostafizur Rahman, Md. Nuruzzaman Khan.

**Writing – original draft:** Mizanur Rahman, Md. Nuruzzaman Khan.

**Writing – review & editing:** Mizanur Rahman, Md Shohel Rana, Gulam Khandaker, Md Mostafizur Rahman, Md. Nuruzzaman Khan.

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
