## [Decision Letter · Decision Letter 0]

17 Sep 2024

PONE-D-24-20059Victimization Status Among Persons with Disabilities and its Predictors: Evidence from Bangladesh National Survey on Persons with DisabilitiesPLOS ONE

Dear Dr. Zaman,

Thank you for submitting your manuscript to PLOS ONE. After careful consideration, we feel that it has merit but does not fully meet PLOS ONE’s publication criteria as it currently stands. Therefore, we invite you to submit a revised version of the manuscript that addresses the points raised during the review process.

We look forward to receiving your revised manuscript.

Kind regards,

Md. Shahjalal

Academic Editor

PLOS ONE

Journal Requirements:

3. Please include captions for your Supporting Information files at the end of your manuscript, and update any in-text citations to match accordingly. Please see our Supporting Information guidelines for more information: http://journals.plos.org/plosone/s/supporting-information .

Reviewers' comments:

Reviewer's Responses to Questions

**Comments to the Author**

1. Is the manuscript technically sound, and do the data support the conclusions?

Reviewer #1: Yes

Reviewer #2: Yes

2. Has the statistical analysis been performed appropriately and rigorously? 

Reviewer #1: Yes

Reviewer #2: Yes

3. Have the authors made all data underlying the findings in their manuscript fully available?

Reviewer #1: Yes

Reviewer #2: Yes

4. Is the manuscript presented in an intelligible fashion and written in standard English?

Reviewer #1: No

Reviewer #2: Yes

5. Review Comments to the Author

Reviewer #1: comments are attached

Abstract:

The following two statements in the same abstract seems contradictory.

“certain divisions like Khulna and Rangpur was associated with lower likelihoods of being victimized” AND

“residence in certain divisions such as Chattogram, Khulna, Mymensingh, Rangpur, and Sylhet reported higher likelihoods of victimization compared to those in the Barishal division”.

Reviewer #2: • Title: The title is clear and concise; it reflects the focus of the study.

• Abstract: The abstract has done a great job summarizing this study, including background, methods, results, and conclusion.

• The writing of the manuscript is good; the logic in the flow from section to section is smooth.

• This study represents a valuable contribution to knowledge about victimization among persons with disabilities in Bangladesh. It carries useful insights that can inform policy and practice.

This paper contributes to victimization among persons with disabilities in Bangladesh with key information in an area that is particularly underresearched, especially in low- and middle-income countries. The study applied a sound methodology by using data from the 2021 National Survey on Persons with Disabilities and multilevel mixed-effect logistic regression analysis to ascertain the crucial predictors of victimization.

Areas of Improvement

Terms and Definitions: There are some terms that, through the manuscript, would have been better defined at the outset. While there was an attempt to explain the outcome variable, a more specific definition of what was meant by "victimization" and its forms, such as psychological, physical, and social, among others, would greatly aid in comprehension. An explanation for why the categories of harassment-for instance, based on ethnicity and religion-were combined into one dichotomous outcome variable would be better stated upfront and transparently.

Discussion of Limitations: Even though the manuscript has stated that this very study is the first of its kind in Bangladesh, an elaboration on limitations was still expected.

For example, self-reported victimization data might be subjected to recall bias or not be reported at all due to the possible stigma that might be associated with them. Specification about how this limitation may have influenced the results could help make this paper even stronger.

Ethical Considerations: Even though the manuscript maintains that no further ethical approval was necessary to undertake the study due to the de-identified nature of the data, it is desired that this section brings out considerations, if any, made to protect the privacy and rights of the respondents while data were being collected.

The focus on the victimization of persons with disabilities is so relevant since this population tends to be marginalized and underrepresented in research. These findings bear crucial implications for policy and strategies of intervention in Bangladesh and other similar contexts.

References

Some references are not updated to reflect the most recent information, such as 2, 22, 31, 33, and 34.

6. PLOS authors have the option to publish the peer review history of their article (what does this mean? ). If published, this will include your full peer review and any attached files.

**Do you want your identity to be public for this peer review?** For information about this choice, including consent withdrawal, please see our Privacy Policy .

Reviewer #1: No

Reviewer #2: **Yes: ** Hala Awad Ahmed

---

## [Author Response · Author response to Decision Letter 1]

29 Sep 2024

We have added a MS word file where we provided point by point response to each reviewer's comments.

---

## [Decision Letter · Decision Letter 1]

23 Oct 2024

PONE-D-24-20059R1Victimization Status Among Persons with Disabilities and its Predictors: Evidence from Bangladesh National Survey on Persons with DisabilitiesPLOS ONE

Dear Dr. Khan,

Thank you for submitting your manuscript to PLOS ONE. After careful consideration, we feel that it has merit but does not fully meet PLOS ONE’s publication criteria as it currently stands. Therefore, we invite you to submit a revised version of the manuscript that addresses the points raised during the review process.

We look forward to receiving your revised manuscript.

Kind regards,

Md. Shahjalal

Academic Editor

PLOS ONE

Reviewers' comments:

Reviewer's Responses to Questions

**Comments to the Author**

1. If the authors have adequately addressed your comments raised in a previous round of review and you feel that this manuscript is now acceptable for publication, you may indicate that here to bypass the “Comments to the Author” section, enter your conflict of interest statement in the “Confidential to Editor” section, and submit your "Accept" recommendation.

Reviewer #1: (No Response)

Reviewer #2: All comments have been addressed

2. Is the manuscript technically sound, and do the data support the conclusions?

Reviewer #1: Partly

Reviewer #2: Yes

3. Has the statistical analysis been performed appropriately and rigorously? 

Reviewer #1: No

Reviewer #2: Yes

4. Have the authors made all data underlying the findings in their manuscript fully available?

Reviewer #1: No

Reviewer #2: Yes

5. Is the manuscript presented in an intelligible fashion and written in standard English?

Reviewer #1: No

Reviewer #2: Yes

6. Review Comments to the Author

Reviewer #1: Major revision. Please submit a revision letter and the manuscript with track changes by addressing all the points raised.

Reviewer #2: Technical Soundness and Data Support:

The research is technically sound and appropriately designed for the question it addresses. The use of multilevel mixed-effect logistic regression is suitable, and the statistical analysis has been rigorously performed. However, improving the clarity of data presentation, particularly in the tables, would strengthen the argument. The conclusions are generally supported by the data, but more detailed consideration of the sample size and limitations is recommended.

Statistical Analysis:

The statistical methods were appropriately chosen and executed. Ensure that any assumptions made during the analysis are clearly described in the methods section for transparency.

Data Availability:

The current manuscript does not meet PLOS ONE’s data availability requirements. It is important to make the data underlying the findings fully accessible or to provide a valid justification if this is not possible. Please update the Data Availability Statement to comply with the journal's policy.

Language and Presentation:

The manuscript is written in clear and standard English. However, there are minor typographical and grammatical errors that should be addressed in the revision process. Enhancing the clarity of complex sections will also improve the manuscript's accessibility to a broader readership.

Ethics and Data Protection:

While the ethics section is present, more detail should be provided about participant consent, data protection measures, and the approval process by the ethics board. This will help meet international publication standards.

Suggestions for Improvement:

Consider including graphical elements (e.g., charts) to enhance the visualization of your findings.

The discussion could benefit from more in-depth reflection on the global relevance of the findings and policy recommendations. Expanding on how these findings compare to other regions or what interventions could be recommended would add practical value.

7. PLOS authors have the option to publish the peer review history of their article (what does this mean? ). If published, this will include your full peer review and any attached files.

**Do you want your identity to be public for this peer review?** For information about this choice, including consent withdrawal, please see our Privacy Policy .

Reviewer #1: No

Reviewer #2: **Yes: ** Hala Awad Ahmed

---

## [Author Response · Author response to Decision Letter 2]

12 Nov 2024

We have uploaded a MS word file where we have provided a point-by-point response to reviewers' comments.

---

## [Editor Report · Decision Letter 2]

2 Jan 2025

Victimization Status Among Persons with Disabilities and its Predictors: Evidence from Bangladesh National Survey on Persons with Disabilities

PONE-D-24-20059R2

Dear Dr. Khan,

We’re pleased to inform you that your manuscript has been judged scientifically suitable for publication and will be formally accepted for publication once it meets all outstanding technical requirements.

Kind regards,

Md. Shahjalal

Academic Editor

PLOS ONE
---

## [Editor Report · Acceptance letter]

PONE-D-24-20059R2

PLOS ONE

Dear Dr. Khan,

I'm pleased to inform you that your manuscript has been deemed suitable for publication in PLOS ONE. Congratulations! Your manuscript is now being handed over to our production team.

Kind regards,

on behalf of

Dr. Md. Shahjalal

Academic Editor

PLOS ONE